# A Facile and Highly Efficient Approach to Obtain a Fluorescent Chromogenic Porous Organic Polymer for Lymphatic Targeting Imaging

**DOI:** 10.3390/molecules27051558

**Published:** 2022-02-25

**Authors:** Man Duan, Dongmei Han, Nan Gao, Wenbin Shen, Kun Chang, Xinyu Wang, Jianshi Du

**Affiliations:** 1Key Laboratory of Lymphatic Surgery Jilin Province, Jilin Engineering Laboratory for Lymphatic Surgery Jilin Province, China-Japan Union Hospital of Jilin University, Changchun 130031, China; duanman19@mails.jlu.edu.cn (M.D.); handm@jlu.edu.cn (D.H.); wangxinyu18@mails.jlu.edu.cn (X.W.); 2Key Laboratory of Polyoxometalate and Reticular Material Chemistry of Ministry of Education and Faculty of Chemistry, Northeast Normal University, Changchun 130024, China; 3Department of Lymphology, Beijing Shijitan Hospital, Capital Medical University, Beijing 100038, China; shenwb@bjsjth.cn (W.S.); chang0213@sina.com (K.C.)

**Keywords:** porous organic polymers, indocyanine green, lymphatic vessel, synergistic effect, π–π interactions

## Abstract

Porous organic polymers have an open architecture, excellent stability, and tunable structural components, revealing great application potential in the field of fluorescence imaging, but this part of the research is still in its infancy. In this study, we aimed to tailor the physical and chemical characteristics of indocyanine green using sulfonic acid groups and conjugated fragments, and prepared amino-grafted porous polymers. The resulting material had excellent solvent and thermal stability, and possessed a relatively large pore structure with a size of 3.4 nm. Based on the synergistic effect of electrostatic bonding and π–π interactions, the fluorescent chromogenic agent, indocyanine green, was tightly incorporated into the pore cavity of POP solids through a one-step immersion method. Accordingly, the fluorescent chromogenic POP demonstrated excellent imaging capabilities in biological experiments. This preparation of fluorescent chromogenic porous organic polymer illustrates a promising application of POP-based solids in both fluorescence imaging and biomedicine applications.

## 1. Introduction

Porous organic polymers (POPs), emerging as a novel functional porous solid, have attracted a great deal of attention due to their open architecture, large surface area, and adjustable pore size [1,2,3]. Thanks to state-of-the-art coupling approaches, great efforts have been made in, for example, crystalline networks of COFs (covalent organic frameworks) and CTFs (covalent triazine-based frameworks) and amorphous networks of PIMs (polymer of intrinsic microporosity), CMPs (conjugated microporous polymers), and PAFs (porous aromatic frameworks) [4,5,6,7,8,9,10,11,12,13,14,15]. In addition to their porous nature, the tunable structural composition of POP samples is a rare quality, which has been widely used to realize custom-made skeletons for the satisfaction of unique requirements in the areas of molecular capture, gas storage, and catalysis [16,17,18,19,20,21,22,23,24,25,26,27]. Although POPs have helped to make great progress in structural design and molecular interaction, their applications in biology are still in their infancy.

Statistical surveys analyzed by the World Health Organization reveal that lymphedema ranks second among disabling diseases, affecting 250 million people worldwide [28]. Near-infrared (NIR) lymphography is an important auxiliary method for the clear diagnosis of lymphedema disease [29,30]. At present, the dye indocyanine green (ICG) is the only clinically approved agent. It is injected into the skin to detect the clinical value of sentinel lymph nodes for various types of cancer (breast cancer, gynecological cancer, and melanoma) [31]. However, ICG has the following shortcomings: (1) The dynamic diameter is relatively small (<1 nm). After subcutaneous injection, ICG quickly enters the capillary veins and lymphatic capillaries, revealing an irregular shape to interfere with the visualization of the lymphatic system [32,33]. (2) Poor physical and chemical properties of ICG, including poor stability, self-quenching, and low quantum yield, seriously affect the application effect during near-infrared lymphography [34,35]. There are many reports of vectors used to encapsulate ICG molecules (e.g., calixarenes) [36,37]. However, their medical fluorescence effect still needs to be improved.

Herein, we prepared an amino-modified POP material using a bottom-up approach. The amino group reacted with the sulfonic acid of indocyanine green through acid–base neutralization, and the aromatic unit in the POP skeleton combined with the conjugated fragment of indocyanine green through π–π interaction. Through the synergy of the electrostatic and π-π interactions, indocyanine green was widely enriched in the porous channels of POP solids. Due to the encapsulation of fluorescent molecules, the fluorescent chromogenic porous organic polymer showed excellent fluorescence imaging capability.

## 2. Results and Discussion

The amino-grafted POP (NH_2_-POP) was synthesized via the palladium/copper-catalyzed Sonogashira–Hagihara cross-coupling reaction using 1,3,5-tris(4-bromophenyl)benzene and 2,5-diethynylaniline as the building monomers (Figure 1) [3]. Then, ICG molecules were incorporated into the porous channels of POP architecture through a one-step immersion process to obtain the fluorescent chromogenic POP product (ICG-POP). 

As shown in Figure 2, FT-IR spectra were analyzed to confirm the preparation of POP samples. The FT-IR spectrum for 2,5-diethynylaniline shows two characteristic signals located at 619 and 672 cm^−1^ ascribed to the asymmetric and symmetric stretching vibrations of the C‒Br bond. As for NH_2_-POP, the C‒Br bond disappears in the IR spectrum, which proves the occurrence of Sonogashira–Hagihara cross-coupling reaction. After loaded with the ICG guest, the appearance of the –S=O signal (1091 and 1421 cm^−^^1^) reveals the successful incorporation of ICG molecules in the POP network. There is a shift of N‒H band from 3379 cm^−^^1^ for NH_2_-POP to 3421 cm^−^^1^ for ICG-POP, manifesting the acid–base neutralization of amino groups with the sulfonic acid groups of IGC. Meanwhile, a series of red shifts including 1457 to 1455 cm^−^^1^, 1468 to 1464 cm^−^^1^, and 1473 to 1471 cm^−^^1^ are attributed to the skeletal vibrations of the phenyl units in the POP skeleton [38,39]. As shown in Appendix A (see in Appendix A), the structural integrity of POP samples was further verified by ^13^C solid-state NMR. The series of peaks in the range of 110–150 ppm are attributed to the aromatic carbons. After adsorption by ICG molecules, the emerging resonance ~50–60 ppm is ascribed to the alkyl carbons of the ICG guest. 

As shown in Figure 3A, there was no obvious characteristic diffraction peak on the XRD pattern, proving that the POP frameworks were short of long-range ordered structure (Figure 3A). As depicted in Figure 3B, there was a small amount of weightlessness in the TG analysis before 100 °C because of the escape of solvent molecules. After that, there was no weight loss before 400 °C, confirming the excellent thermostability of NH_2_-POP and ICG-POP. Finally, no residues for either NH_2_-POP or ICG-POP were left after burning to 800 °C, which is indicative of no catalyst residue in the POP samples. 

Nitrogen sorption experiments were conducted at 77 K to investigate the porosity of as-prepared POP materials. As illustrated in Figure 3C, a sharp increase in gas uptake was observed at low pressure in the nitrogen adsorption–desorption isotherms of NH_2_-POP and ICG-POP, indicating the existence of micropores in both POPs. The BET surface area of NH_2_-POP was 659 cm^2^ g^−^^1^. The corresponding pore sizes were 1.4 and 3.5 nm for ICG-POP and NH_2_-POP, respectively, determined on the basis of non-local density functional theory (Figure 3D). This large pore space was conducive to the storage of ICG molecules in the porous channels. After loading, the BET surface area of ICG-POP was 533 cm^2^ g^−^^1^. The decrease in surface area is attributed to the occupancy of ICG molecules in the porous channels, which increased the weight per structural unit and decreased the adsorption capacity of N_2_ gas molecules. 

The morphology of POP solids was monitored by scanning electron microscopy (SEM), which showed that both NH_2_-POP and ICG-POP were aggregates of micro-particles with a size of 800 nm. The appearance of ICG-POP was similar to that of NH_2_-POP, indicating that ICG molecules were wrapped in the pore channels (Figure 4). Determined on the basis of non-local density functional theory, the pore sizes of NH_2_-POP were concentrated at 1.2 and 3.5 nm, respectively. The mesoporous channels with a size of 3.5 nm provided a large space for the rapid transport of indocyanine green molecules inside the POP particle, while the 1.2 nm microporous cavities could accommodate indocyanine green molecules with high capacity and strong binding force through the action of van der Waals force. After loading with indocyanine green molecules, the pore sizes of the ICG-POP were still concentrated at 1.2 and 3.5 nm. As for pore volume, the pore volume of the 3.5 nm pore size did not change significantly, while that of the 1.2 nm pore size was significantly reduced. This conclusion also proves our speculation that after the indocyanine green molecules enter the interior of the NH_2_-POP, they quickly fill the 1.2 nm micropores. According to Beer–Lambert’s law, the highest ICG uptake of ICG-POP was calculated to be ~128 mg g^−1^ in 90 min (Appendix A).

The lymphatic system is vitally important for the body’s defense, and can effectively remove foreign bodies and bacteria from the living organism [40,41,42,43]. Lymph nodes are an important functional element of the lymphatic system, meanwhile, they are the reservoir for the root and seed of many cancer cells in the human body [44,45]. Therefore, the identification of different types of cells in lymph nodes is an efficient and effective way to pre-evaluate cancer diseases. To date, ICG is the only drug approved and certified by the FDA for lymphatic vessels imaging. However, because of the small particle size, indocyanine green is easily degraded in vivo, and has no selective and rapid response ability in lymph nodes. To address this issue, fluorescent porous organic polymers have been applied in the cell imaging field. 

For lymphatic targeting, it is widely recognized that the size of materials determines their selective entrance into blood or lymphatic vessels. Typically, nanoparticles 5–100 nm in diameter are suitable for lymphatic drainage, given the maximum gap (ca. 120 nm) between the endothelial cells of the lymphatic capillary. In addition to size, surface charge also has a great influence on delivery efficiency. Although there remain some disputes in this regard, hyaluronic acid (HA) with negative charge plays a positive role in the lymphatic targeting effect. Besides the factors mentioned above, the sites of subcutaneous injection also play an important role in lymphatic targeting [46]. In order to realize the targeted fluorescence recognition of the cancer cell, we first modified the surface of the polymer particles with hyaluronic acid through esterification reaction, and the ICG uptake of ICG-POP-HA was calculated to be ~122 mg g^−1^ (Appendix A). 

The potential toxicity of ICG-POP was tested by MTT assay. As shown in Appendix A, the proliferation rate of cells treated with ICG-POP indicated the excellent biocompatibility of ICG-POP. Normal lymphatic endothelial cells (MLEC), murine macrophage cells (RAW264.7), and mouse colon cancer cells (CT26) were utilized to examine the fluorescence imaging properties of POP solids. The three types of cells were placed in Petri dishes with a 1 cm × 1 cm glass sheet for 24 h to obtain a glass sheet containing cells. Then the glass containing cells was processed with the fluorescent ICG-POP material. Finally, the treated glass slides were washed with pure saline several times, after which the cells were settled on the glass by 4% paraformaldehyde for further analysis.

After imaging by confocal laser scanning microscopy, Figure 5 shows the staining effect of cells with only indocyanine green molecules (Figure 5A) and fluorescent ICG-POP (Figure 5B). As shown in Figure 5A, the indocyanine green molecules were evenly dispersed in the three cells, neither selectively nor specifically binding to cancer cells. Treated with fluorescent ICG-POP, only the cancer cells showed significantly enhanced fluorescence property. This is attributed to the fact that the strong binding affinity between LYVE-1 and HA units brings about rapid migration and long retention, leading to high performance for the lymphatic vessel imaging of ICG-POP.

To further test the applicability of ICG-POP, mouse leg tissues were selected. After staining with ICG-POP, LYVE-1 antibody, or both, it is possible to clearly distinguish the overlaid image treated by both ICG-POP and LYVE-1 antibody, illustrating the excellent performance for imaging within lymphatic vessels (Figure 6).

## 3. Materials and Methods

### 3.1. Chemicals

All chemicals were purchased from commercial suppliers and used as received unless noted otherwise. *N*,*N*-Dimethylformamide (DMF) and triethylamine (Et_3_N) were dehydrated with Mg_2_SO_4_. Tetrahydrofuran (THF) was distilled in the presence of sodium benzophenone ketyl under N_2_ atmosphere.

### 3.2. Synthesis of Amino-Grafted POP (NH_2_-POP)

1,3,5-Tris(4-bromophenyl)benzene (1086 mg, 2 mmol), 2,5-diethynylaniline (423 mg, 3 mmol), tetrakis-(triphenphosphine)palladium (60 mg), and cuprous iodide (16 mg) were dissolved in a mixture of DMF (10 mL) and Et_3_N (10 mL) in a 50 mL two-necked flask. After degassing via three freeze–pump–thaw cycles, the mixture was stirred at 85 °C for 72 h under N_2_ atmosphere. After cooling to room temperature, the resulting precipitate was collected by filtration, followed by consecutive washing by Soxhlet extraction for 72 h with THF, methanol, and dichloromethane to remove the unreacted monomers or catalysts. After drying at 100 °C in vacuum for 12 h, a brown powder was collected and named as NH_2_-POP (720 mg, 80% yield).

### 3.3. Synthesis of Fluorescent Chromogenic POP (ICG-POP)

An amount of 25 mg of indocyanine green was dissolved in 10 mL of ethanol, and then the solution was added to the ethanol solution of POP powder (40 mg). The reaction system was stirred for 12 h, and the insoluble solids were filtered out by vacuum filtration. After rinsing with water, acetone, and dichloromethane, respectively, for 10 min, the indocyanine green-doped POP material was obtained.

### 3.4. Characterization

Fourier-transform infrared spectroscopy (FT-IR) was conducted on a Nicolet Impact 410. Thermogravimetric analysis (TG) was conducted using a Netzch Sta 449c thermal analyzer with a heating rate of 10 °C min^−^^1^ under air conditions. Scanning electron microscopy (SEM) imaging and energy-dispersive spectroscopy (EDS) were performed on a JEOS JSM 6700. N_2_ adsorption–desorption measurement was analyzed on a Quantachrome Autosorb-IQ gas adsorption analyzer.

### 3.5. Cellular Accumulation and Quantification of the Fluorescence Intensity

MLEC cells, RAW 264.7 cells, and CT26 cells were seeded on six-well plates at 1 × 10^5^ cells per well. After 12 h incubation, cells were treated with ICG or ICG-POP. Then, cells were washed between 1 and 5 times with PBS and incubated in fresh medium. The fluorescence of cells was estimated by CLSM.

## 4. Conclusions

In this paper, a porous organic polymer was synthesized by using Sonogashira–Hagihara cross-coupling reaction which contained many amino groups on the pore surface. Based on the synergistic effect of electrostatic bonding and π–π interactions, the fluorescent chromogenic agent indocyanine green was tightly incorporated into the pore cavity of POP solids through a one-step immersion method. Through experiments using various cell lines and mouse tissues, the fluorescent chromogenic POP revealed excellent imaging capabilities in biological experiments. Our work not only provides a simple method for the preparation of fluorescent porous materials, but also paves the way for the design of biodegradable nano-tracers.

## Figures and Tables

**Figure 1 molecules-27-01558-f001:**
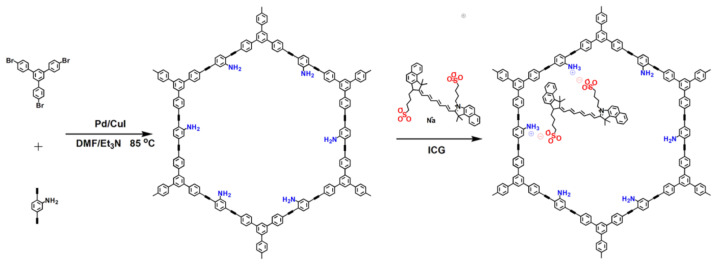
Synthesis of NH_2_-POP and ICG-POP [3].

**Figure 2 molecules-27-01558-f002:**
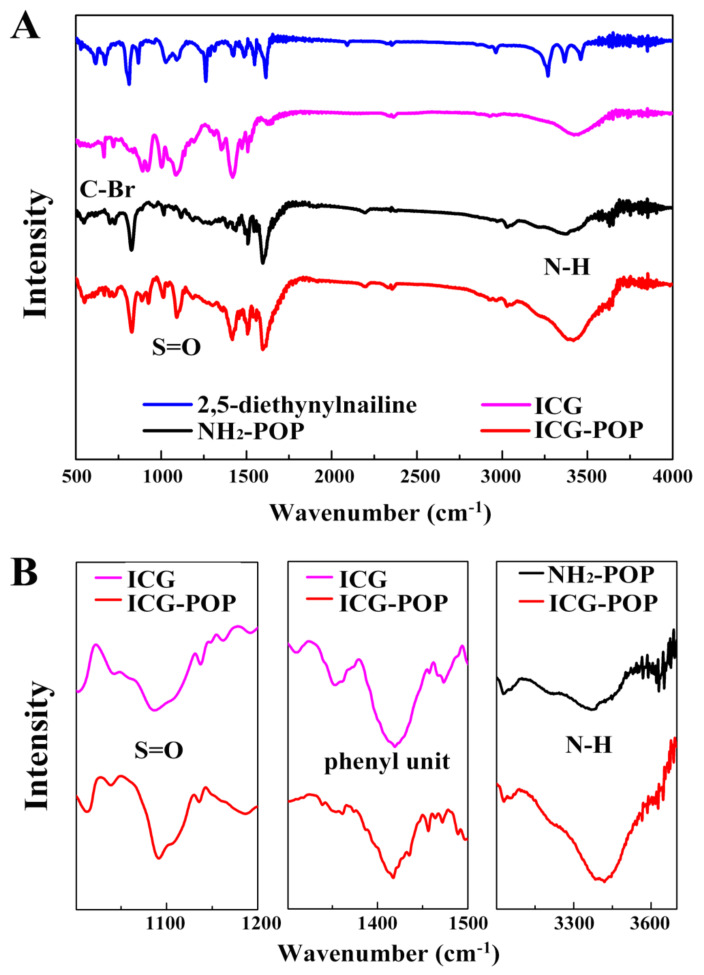
FT-IR spectra of 2,5-diethynylaniline, ICG, NH_2_-POP, and ICG-POP (**A**), and enlarged spectra (**B**).

**Figure 3 molecules-27-01558-f003:**
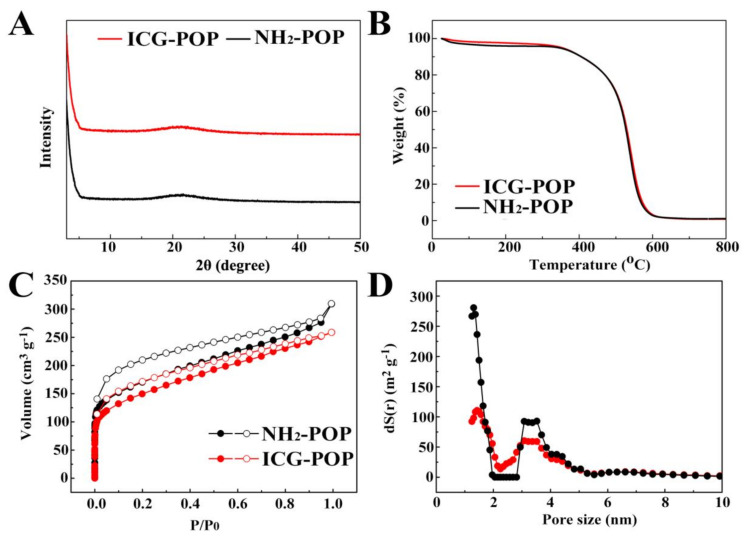
The PXRD patterns (**A**), TGA curves (**B**), N_2_ adsorption–desorption isotherms (**C**), and pore size distribution (**D**) of NH_2_-POP and ICG-POP.

**Figure 4 molecules-27-01558-f004:**
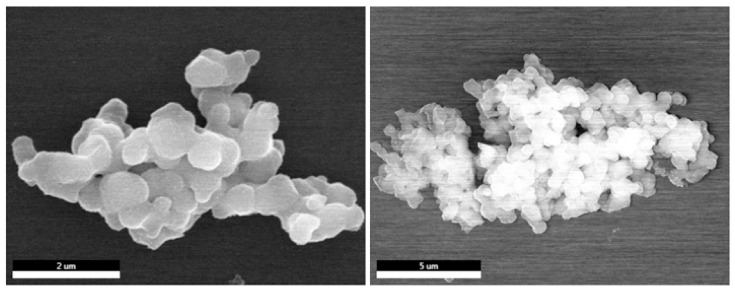
SEM images for NH_2_-POP (**left**) and ICG-POP (**right**).

**Figure 5 molecules-27-01558-f005:**
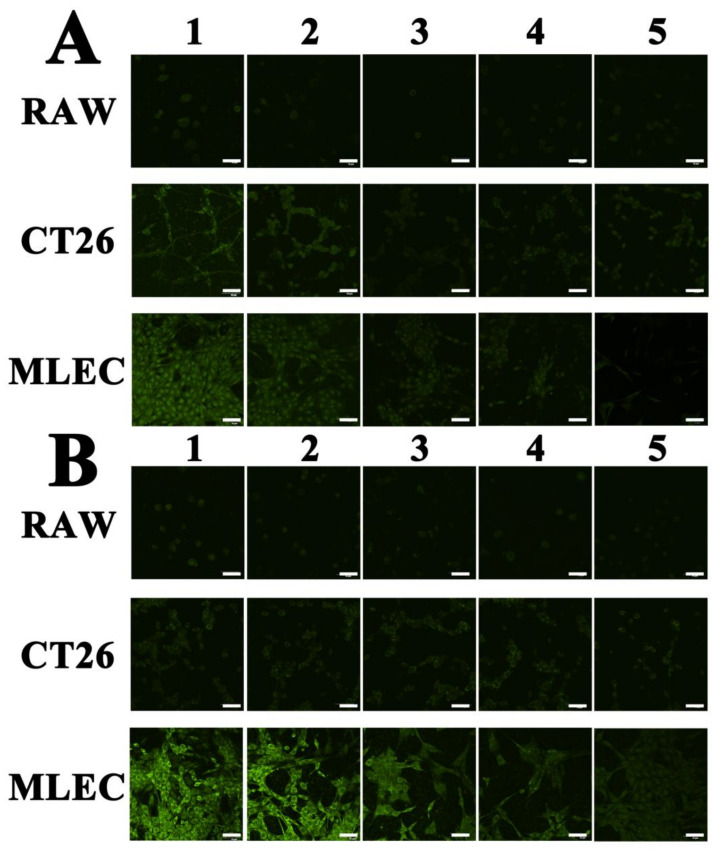
The staining effect in cells by ICG (**A**) and ICG-POP (**B**). The numbers (1, 2, 3, 4, and 5) represent the number of times cells were washed with pure saline after staining. The scale bars equal 50 μm.

**Figure 6 molecules-27-01558-f006:**

Mouse tissues stained with respective ICG-POP (**A**), LYVE-1 antibody (**B**), or both (**C**). The scale bars equal 200 μm.

## Data Availability

All data related to this study are presented in this publication.

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
