# Peer review of "A Facile and Highly Efficient Approach to Obtain a Fluorescent Chromogenic Porous Organic Polymer for Lymphatic Targeting Imaging"

_molecules, 2022, doi:10.3390/molecules27051558_

Round 1
Reviewer 1 Report
All my comments and concerns have been addressed, so I recommendation of the manuscript for publication.
Author Response
Please see the attach.

Reviewer 2 Report
This manuscript introduces a novel idea of exploiting a porous organic polymer in order to to obtain a new biodegradable nano-tracer that can be v efficient in fluorescence bioimaging.
The manuscript is well-designed but needs thorough revision of its English language regarding the grammar and the spelling of some words such "copper". Please also re-phrase the paragraph in the introduction starting with: "Gathered by the WHO...." and put the word "the" before the word "dye" in the first sentence in page 2.
Author Response
Please see the attach.

Reviewer 3 Report
The paper reports on the use of porous organic polymers with indocyanine green incorporated to target the lymphatic nodes.
The work shows some serious flaws in the planning, execution and presentation of the experimental work. The overall quality of the manuscript is far from the required standard for publication in Molecules.
The structural and some morphological characterization is shown for NH2-POP and ICG-POP, but no structural nor morphological characterization is presented for the actual ICG-POP-HA material that was tested in cells and in mouse tissue. A discussion of the optical properties of the ICG-POP-HA material that is supposed to be used as a fluorescent label of the lymphatic nodes in also missing. The emission yield of the incorporated dye should be provided, or at least it should be demonstrated that the dye maintains its optical properties when incorporated in the POP. There is a serious possibility the π-π staking to the walls of the porous structures could lead to the quenching of emission. The amount of loading of ICG on the ICG-POP-HA is not known.
Critical experimental details are missing to ensure a correct interpretation of the data and also reproducibility of the procedure. The details about HA functionalization are missing. The experimental details about the 2D cell culture experiments are incomplete. Details such as cell density, incubation media, concentration of the POPs in the incubation media and incubation time are missing. No information whatsoever is provided about the staining of the mouse tissue. The details of the imaging acquisition are also missing. The rational behind the biocompatibility studies in only one of the cell models is unclear.
The discussion lacks an in-depth interpretation of the data. The confocal imaging data is discussed only qualitatively. The interpretation of the FTIR spectra has gross mistakes, such as the identification of C-Br stretching vibrations in the spectrum of a molecule that does not have any bromine atom. The existence of pores of two different sizes from the adsorption desorption experiments is not discussed. The XRD diffraction shows a broad band at 20º that is completely overlooked.
Finally, the introduction fails to describe other relevant strategies to target lymphatic nodes. Many data is provided in the supporting information that is not mentioned in the manuscript.
Author Response
Please see the attach.

Round 2
Reviewer 3 Report
The authors have included experimental details that were missing but did not made a real effort to improve the overall quality of the manuscript . The introduction of the experimental details and critical experimental data in the supporting information has not been reflected in a more accurate presentation of the critical data, nor in a more in-depth discussion of the data, or better supported conclusions in the manuscript.
The introduction still fails to describe other relevant strategies to target lymphatic nodes. Many relevant data is provided in the supporting information that is not discussed or even mentioned in the manuscript. There is a general lack of accuracy in the presentation and discussion of the data. My overall assessment continues to be that the quality of the manuscript is far from the required standard for publication in Molecules.
As an example, it is positive that the authors introduced figure S10 in the supporting information, however the data in the figure does not seem to fully support the coclusions in the manuscript. The figure itself is not even discussed in the text. From this figure we clearly see that both ICG and ICG-POP have enhanced intensity in MLEC cells. Thus, the results do not support the conclusion in the manuscript that ICG is evenly dispersed in the three cells, nor that for ICG-POP only the cancer cells showed significantly enhanced fluorescence, as stated in page 5. In Fig S10 the caption is not accurate, figure shows plots of intensity distribution with indication of average intensity. The axis in the plots are not correctly identified, nor it is clear that the values in the figure correspond to average intensity.
The authors have included information about ICG uptake in figure S3, but no information is provided as to how this uptake was evaluated. The interpretation of figures S2 and S3 is unclear. This is citical data that should be in the manuscript.
In figure S7 the caption and the legend of the figure are contradictory. In the caption there is the black trace is identified as NH2-POP, whereas in the figure it is identified as ICG.
In figure S8 there should be an indication of the concentration of the solutions.
In the FTIR, how is it possible that the 2,5-diethynylaniline shows two characteristic signals of stretching vibrations of C‒Br bond.
This manuscript is a resubmission of an earlier submission. The following is a list of the peer review reports and author responses from that submission.
Round 1
Reviewer 1 Report
This paper describes the synthesis and characterisation of a new porous organic polymer (POP). The reported POP exhibits imaging properties after encapsulating the fluorescent chromogenic agent, indocyanine green. This is an interesting piece of work; however, although the biological properties have been well studied, the structure of the POP and its adsorption capability need to be further supported. 1H-NMR studies are essential to confirm the structure and the purity of the POP. Thermodynamic and kinetic studies can be performed for the encapsulation to reveal the maximum encapsulation capacity and provide insights in to the adsorption mechanism.
Reviewer 2 Report
This manuscript from Duan and coworkers describes the synthesis of a porous organic polymer (NH2POP), based on the coupling of tris(bromomethyl)benzene with a dialkynylaniline, and the formation of an inclusion compound between the POP and indocyanine green (ICG). The thus formed material (ICGPOP) was partially characterized (IR, XRD, TG, SEM, N2 absorption) and used in cell imaging over three different cell lines (MLEC, RAW264.7 and CT26). While the topic presented shows, in principle, some interest, the manuscript is not acceptable in its present form because it contains several troublesome aspects that must be solved before publication. I recommend rejection and resubmission.
The first aspect is the characterization of both the free material (NH2POP) as well as the inclusion compound ICGPOP. NMR data must be provided for NH2POP to check that the reaction happened as the authors show in Figure 1. It seems that NH2POP is a new compound, because the authors do not mention a previous synthesis of this compound. Then, NMR data are necessary to properly characterize it (in solution or even in solid). If NMR data can't be achieved, neither in solution nor in solid, the authors have to explain why. In addition, molecular composition and purity must be confirmed using microanalytical data (CHN) or high resolution mass spectrometry (HRMS). This is necessary, as this will avoid sentences as "which contains a lot of amino groups in the pore surface" (conclusions). For the ICGPOP material the situation is even worse. The authors detail that the compound ICGPOP has "excellent solvent and thermal stability", therefore I don't see reasons to elude a classical organic characterization in solution (NMR + HRMS). In the case of ICGPOP the authors propose the inclusion of one molecule of ICG every 6 NH2 groups (Figure 1), but no proofs at all (of any kind) are given to support this formula throughout the text. Why this relationship? Is it possible another different molar ratio? The lack of characterization involves also the fluorescence measurement. The authors argue that the inclusion of ICG into POP improves the fluorescence, but no measurements of the fluorescence of these materials have been presented, no values of quantum yields are given. A comparison of the fluorescence of free ICG and that of ICGPOP is mandatory to see the true effect of the inclusion in the POP.
The second aspect is the lack of context of the research. There are examples of ICG supported in other materials (for instance, calixarenes, medchemcomm 2016, 7, 623), and even in polymers, and this should mentioned to understand the progress here reported. The materials here reported and their properties has to be compared with other materials and their respective photophysical properties. Otherwise the synthesis has not sense. For comparative purposes check ChemPhotoChem 2021, 5, 727.
In close relationship with the previous point, the bibliography presented in this manuscript has also to be updated. Being this area in constant evolution, references 1-15 should be present revision works, and not puntual contributions which in most of the cases are from 2005-2008. For instance, references 1-3 dealing with POP can be updated with Mat. Chem. Front. 2020, 4, 332, and so on for references 4-15. Especially representative is reference 18, talking about ranking of lymphedema, which is from 1969 (!). Please check J. Am. Acad. Dermatol. 2017, 77, 1009.
Reviewer 3 Report
This manuscript reported the synthesis of amnio-grafted porous organic polymers (NH2-POP) for lymphatic targeting imaging in biomedicine applications. The authors described the synthesis procedures and mechanisms, conducted some basic material characterization measurements, and compared the staining effect of cells by dye indocyanine green (ICG) with and without NH2-POP. Overall, this is an interesting work and deserves publication in molecules. Therefore, I suggest a minor revision before accepting for publication, and my comments are as follows:
- The font of the works in Fig. 1 should be enlarged to make them readable for readers.
- Proper reference should be provided to support the reaction mechanism in Fig. 1.
- The FTIR spectra in Fig. 2 should be indexed with the corresponding components according to the discussion on Page 2.
- The discussion of the PXRD patterns in Fig. 3a is not sufficient. How did the authors draw the conclusion that the POP frameworks were short of long-range ordered structure? More detailed discussion and references are needed.
- For the Materials and Methods section, the purity and provider information of the chemicals should be provided along with the synthesis procedure.
- Section 3.4, the description of XRD measurements should be provided
Round 2
Reviewer 1 Report
The authors have addressed all the comments addressed by the referees and the manuscript is now suitable for publication in Molecules without further alternations.